# Trimethylamine N-Oxide as a Mediator Linking Peripheral to Central Inflammation: An In Vitro Study

**DOI:** 10.3390/ijms242417557

**Published:** 2023-12-16

**Authors:** Manuel H. Janeiro, Maite Solas, Josune Orbe, Jose A. Rodríguez, Leyre Sanchez de Muniain, Paula Escalada, Ping K. Yip, Maria J. Ramirez

**Affiliations:** 1Department of Pharmacology and Toxicology, University of Navarra, 31008 Pamplona, Spain; mhjaneiro@alumni.unav.es (M.H.J.); msolaszu@unav.es (M.S.); lsanchezdem@alumni.unav.es (L.S.d.M.); pescalada@alumni.unav.es (P.E.); 2IdISNA, Navarra Institute for Health Research, 31008 Pamplona, Spain; 3Laboratory of Atherothrombosis, CIMA, 31008 Pamplona, Spain; josuneor@unav.es; 4Redes de Investigación Cooperativa Orientadas a Resultados en Salud (RICORS)-Ictus, Instituto de Salud Carlos III, 28029 Madrid, Spain; 5CIBER Cardiovascular (CIBERCV), Instituto de Salud Carlos III, 28029 Madrid, Spain; 6Centre for Neuroscience, Surgery & Trauma, Blizard Institute, Queen Mary University of London, London EC1M 6BQ, UK; p.yip@qmul.ac.uk

**Keywords:** cytokines, blood–brain barrier, adipocytes, macrophages, microglia

## Abstract

In this study, the plausible role of trimethylamine N-oxide (TMAO), a microbiota metabolite, was investigated as a link between peripheral inflammation and the inflammation of the central nervous system using different cell lines. TMAO treatment favored the differentiation of adipocytes from preadipocytes (3T3-L1 cell line). In macrophages (RAW 264.7 cell line), which infiltrate adipose tissue in obesity, TMAO increased the expression of pro-inflammatory cytokines. The treatment with 200 μM of TMAO seemed to disrupt the blood–brain barrier as it induced a significant decrease in the expression of occludin in hCMECs. TMAO also increased the expression of pro-inflammatory cytokines in primary neuronal cultures, induced a pro-inflammatory state in primary microglial cultures, and promoted phagocytosis. Data obtained from this project suggest that microbial dysbiosis and increased TMAO secretion could be a key link between peripheral and central inflammation. Thus, TMAO-decreasing compounds may be a promising therapeutic strategy for neurodegenerative diseases.

## 1. Introduction

Trimethylamine N-oxide, also known as TMAO, is a molecule generated by gut microbiota from substrates such as L-carnitine and choline. These substrates are converted to TMA by numerous enzymes and then transformed into TMAO by the hepatic enzymes flavin-containing monooxygenase 1 (FMO1) and flavin-containing monooxygenase 3 (FMO3) [1].

As TMAO proceeds from choline compounds such as phospholipids, it has been suggested to be augmented after high-fat diet consumption [2,3,4,5]. Moreover, some studies have also associated TMAO with obesity [6]. At first, TMAO was thought to be a waste product with no action in the organism; however, nowadays there is emerging evidence describing TMAO as a proatherogenic molecule closely related to inflammation and aging [3,7,8,9,10,11,12,13]. Aging is characterized by changes in gut microbiota composition decreasing its diversity, which could play a key role in the development of several disturbances such as insulin resistance and even neuropathologies [1]. Dysbiosis is able to reduce the expression of tight junction proteins in the gut, inducing intestinal inflammation and a major permeability, which favors the passage of gut metabolites to the circulatory system [1]. Among these microbial metabolites, trimethylamine N-oxide (TMAO) has been related to inflammation and cognitive decline [1].

Moreover, TMAO has been associated with major production of pro-inflammatory cytokines and increased expression in macrophages of scavenger receptors CD36 and SR-A1, which promote lipid accumulation and foam cell formation [14,15,16].

All this evidence would lead to peripheral inflammation. However, there is scarce evidence linking TMAO and neuroinflammation. Some studies have suggested that TMAO crosses the blood–brain barrier (BBB) easily reaching the brain [17,18,19]. One of the main structural features of the BBB is the presence of tight junctions formed by transmembrane proteins, such as occludin, junctional adhesion molecules, and claudins that interact with tight junction-associated proteins, such as the scaffolding proteins zonula occludens (ZO) 1 or 2. Interestingly, recent studies have suggested that TMAO [20,21] or even pro-inflammatory cytokines [22,23] are able to impair the BBB by reducing tight junction proteins, which facilitates their entrance as well as other inflammatory molecules into the brain.

On the other hand, fish are rich in TMAO and choline compounds, and their consumption is widely related to a healthy lifestyle and less inflammation. Moreover, TMAO has been reported to have a protective role in marine organisms as it acts as a chaperone stabilizing proteins from changes in pH and osmotic and hydrostatic stress [1]. In addition, some in vitro experiments have also reported that TMAO could give back the primary structure of unfolded proteins, recovering its function.

Whether TMAO is beneficial or harmful remains unknown and its mechanism in inflammation has not been fully elucidated yet but NLRP3 inflammasome seems to be involved somehow [7,9,10,21,24,25,26]. On the other hand, some studies have suggested that the MAPK/JNK pathway plays a critical role in TMAO-induced atherosclerosis [27].

Hence, in this study, we performed in vitro pharmacological and toxicological studies with TMAO to try to understand its role in the organism. As TMAO may be related to obesity, the role that TMAO could play in adipocytes (3T3L-1 cell line) is first studied. Then, the role TMAO may exert in macrophages is explored as these may infiltrate into adipose tissue and release pro-inflammatory cytokines.

These pro-inflammatory markers could lead to peripheral chronic inflammation and impair the BBB, reducing tight junction protein expression; however, in the present work, the role TMAO could play in the BBB per se is also assessed. Several studies have shown that TMAO can cross the BBB; therefore, the interaction between TMAO and neurons is also investigated. Finally, neuroinflammation is mainly driven by astrocytes and microglia; thus, the present study covers the role of TMAO in microglia activation and phagocytosis.

## 2. Results

### 2.1. TMAO Favored Differentiation of Adipocytes from Preadipocytes (3T3-L1)

An MTT assay showed that increasing TMAO concentrations did not negatively affect cell viability in adipocytes until it reached the concentration of 1000 μM (Figure 1A) (F = 4.532, *p* < 0.001). Using the range of concentration that does not induce cell death, TMAO was associated with a major differentiation rate of mature adipocytes from preadipocytes. As shown in Figure 1B, the percentage of differentiated mature adipocytes augmented above 35% when TMAO 200 μM (135.8 ± 4.946) or 400 μM (136.9 ± 6.965) was added to the medium (F = 5.417, *p* < 0.01). The addition of TMAO 50 μM showed a tendency toward a major differentiation rate but this tendency was not significant. Thus, TMAO did not negatively affect 3T3-L1 but significantly increased their proliferation vs. the control.

### 2.2. TMAO Increased Expression of Pro-Inflammatory Cytokines in Macrophages (RAW 264.7)

Given the demonstrated relationship between TMAO and inflammation, we studied the expressions of pro-inflammatory markers such us IL-6, IL-1β, and TNF-β in macrophages as they have been shown to infiltrate the adipose tissue and cause inflammation. Once again, an MTT assay was performed in order to determine the range of concentrations that do not lead to cell death; it was found that only 1000 μM decreased cell viability (Figure 2A) (F = 4.410, *p* < 0.0001). Interestingly, the expressions of pro-inflammatory cytokines IL-1β (Figure 2B) (F = 6.608, *p* < 0.05), IL-6 (Figure 2C) (F = 16.35, *p* < 0.001), and TNF-α (Figure 2D) (F = 9.207, *p* < 0.01) were found to be augmented after treatment with TMAO in the amounts of 200 µM and 400 µM.

### 2.3. TMAO Decreased Expression of Tight Junction Proteins in hCMECs

The need for a human BBB model that closely mimics the in vivo phenotype and is reproducible and easy to grow has been widely recognized by cerebrovascular researchers. While primary human cerebral microvessel endothelial cells would ideally be the model of choice, the paucity of available fresh human cerebral tissue makes wide-scale studies impractical. Human cardiac microvascular endothelial cells (hCMECs) represent one such model of the human BBB that can be easily grown and is amenable to cellular and molecular studies [28]. As depicted in Figure 3A, only the treatment with 1000 μM of TMAO was able to decrease hCMEC viability (F = 9.949, *p* < 0.0001). The treatment with 200 μM of TMAO induced a significant decrease in the expression of occludin (Figure 3B) (Student’s *t*-test, *p* < 0.05) and a strong tendency toward a decrease in ZO-1 (Figure 3C) (Student’s *t*-test, *p* = 0.06).

### 2.4. TMAO Increased Expression of Pro-Inflammatory Cytokines in Primary Neuronal Cultures

In parallel with the observations in other cell types, only 1000 μM of TMAO decreased primary neuronal cell culture viability (Figure 4A) (F = 3.330, *p* < 0.01). Moreover, TMAO treatment induced a significant increase in the expressions of IL-1β (Figure 4B) (F = 10.89, *p* < 0.01 with 200 and 400 μM of TMAO) and IL-6 (Figure 4C with 400 μM of TMAO) (F = 6.379, *p* < 0.05) in primary neuronal cultures. Unexpectedly, no significant changes were observed in TNF-α expression (Figure 4D) (F = 3.002, *p* = 0.10).

### 2.5. TMAO Induces a Pro-Inflammatory State in Primary Microglial Cultures and Promotes Phagocytosis

TMAO treatment induced an increase in the pro-inflammatory marker CD16/CD32 at all concentrations studied, suggesting a microglial activation toward a pro-inflammatory phenotype (Figure 5A,B) (F = 2.874, *p* < 0.05). Moreover, TMAO significantly increased myelin phagocytosis in primary microglial cell cultures (Figure 5C,D) (F = 11.14, *p* < 0.0001).

## 3. Discussion

TMAO has recently emerged as a pro-inflammatory molecule related to atherosclerosis, which may also be involved in neurological pathologies. This molecule proceeds from the bacterial synthesis of common substrates found in our diets such as L-carnitine and choline. Noteworthy, some risk factors like aging, disease, or diet may lead to gut dysbiosis, thus favoring TMA-forming microbiota, leading to higher TMAO levels after its conversion in the liver.

The consumption of diets rich in both fat and sugar has been observed to impact the composition of the microbiota, potentially causing an imbalance in the gut’s microbial environment and the integrity of the intestinal wall, which causes the translocation of bacterial lipopolysaccharides (LPSs) into the blood [29]. This elicits the secretion of pro-inflammatory cytokines, resulting in inflammation. As such, a recent hypothesis postulates that the gut microbiota may serve as a mechanistic bridge linking the consumption of imbalanced diets, particularly those high in fat, with compromised cognition [30,31]. The literature illustrates that even brief exposure to specific diets can rapidly alter the composition of the microbiota in both human subjects [32] and mice [33], potentially influencing the early onset of cognitive decline [34]. Indeed, preclinical experiments validate that a high-fat diet (HFD) can modify the gut microbiota, potentially contributing to the development of dementia [35,36,37,38]. Intriguingly, a recent study investigating microbiome–metabolome patterns in a mouse model of Alzheimer’s disease (AD), subjected to either a standard or fatty diet, illuminated striking similarities in gut microbiome abnormalities between high-fat feeding and a genetic predisposition to neurodegenerative disorders [38]. Through the analysis of serum and fecal metabolites, this study revealed deficiencies in unsaturated fatty acids and choline, alongside excesses in ketone bodies, lactate, amino acids, trimethylamine (TMA), and trimethylamine N-oxide (TMAO) in AD mice fed a fatty diet. These metabolic imbalances are, in turn, closely associated with cognitive impairment and cerebral hypometabolism [1].

Based on this evidence and in order to better understand if TMAO could be the link between metabolic diseases such as obesity, we started studying the adipocyte (3T3-L1 cell line) cell viability upon a TMAO treatment. Surprisingly, cell viability seemed not to be negatively affected by TMAO. On the contrary, TMAO treatment favored survival over the control. This was further confirmed via a cell proliferation assay, where TMAO increased the differentiation of mature adipocytes from preadipocytes, thus favoring the increase in fat adipose tissue. These results are in line with other studies showing that TMAO levels are linked to adiposity and could increase the expressions of scavenger receptors CD36 and SR-A1 in macrophages [14,15,16]. These scavengers promote lipid accumulation and foam cell formation. Moreover, other studies have shown that FMO3 expression (the enzyme that converts TMA into TMAO) is closely related to fat adipose tissue and its inhibition converts WAT adipocytes to “brown-like” adipocytes known as beige cells that promote protection against obesity [6].

The ingress of macrophages into adipose tissue plays a pivotal role in amplifying the expression of inflammatory cytokines in the context of obesity. However, the underlying causes behind the intensified infiltration of macrophages into adipose tissue and the precise signals that trigger the release of pro-inflammatory cytokines remain undefined. Multiple investigations have shed light on the escalation in pro-inflammatory cytokine expression, aligning with heightened plasma levels of trimethylamine N-oxide (TMAO). A study conducted by Rohrmann et al. (2016) [8] unveiled a link between low-grade inflammation and increased plasma TMAO concentrations. Notably, an elevation in plasma TMAO concentration correlated with an upregulation of TNF-α, IL-6, and C-reactive protein expressions. In our hands, TMAO 200 μM was able to increase the expressions of pro-inflammatory cytokines IL-6, TNF-α, and IL-1β. These results agree with the results shown previously by Geng et al. [27], who showed a huge rise in protein and mRNA expressions of IL-6, TNF-α, and ICAM-1 in macrophages after TMAO treatment. The concentration used for the experiments was carefully determined via the cell viability assay, which demonstrated that the only concentration that affected cell viability was 1000 μM. Based on this evidence and on the concentration of TMAO used for in vitro experiments in the literature for other cell lines [7,9,10,16,21,27,39,40], all of our experiments included the concentration of 200 μM because recent epidemiological studies have shown that TMAO in plasma ranges widely from 0.08 to 250 μM [41].

TMAO has been suggested to cause BBB disruption by reducing the expressions of tight junction proteins like occludin and tight junction protein-1 (ZO-1), favoring its access to the brain [20,21]. In our hands, TMAO concentrations that did not alter cell viability (under 1000 µM) were able to reduce the expressions of tight junction proteins like zonulin and occludin in hCMECs, suggesting that if this occurs in vivo it may induce a higher BBB leaking. In this condition, aberrant amounts of plasma pro-inflammatory cytokines and other detrimental molecules (such as TMAO per se) may enter the brain via the BBB and cause inflammation by altering microglial maturation [42] and astrocyte activation [43].

On the other hand, recent studies have suggested that TMAO can reach the brain and cross the BBB and/or the blood–CSF barrier [18,19]. As TMAO seems to be able to cross the BBB, it may have direct contact with brain cells such as neurons, astrocytes, and microglia. Cell viability of primary neuronal cultures was not affected by TMAO. However, it raised the expression of pro-inflammatory cytokine IL-1β. This result is also in line with other evidence in the literature showing that NLRP3 inflammasome is a cytokine-activating protein complex of the IL-1β family [26] that has been widely linked to TMAO [7,9,10,21,24,25,26]. Thus, NLRP3 formation and activation would cause a major expression and release of the pro-inflammatory cytokine IL-1β [9,21] that would lead to neuroinflammation. In recent years, it has become very clear that neuroinflammation has a central role in the pathogenesis of CNS-related diseases, and thus targeting the regulation or availability of inflammatory mediators can be used as a therapeutic target. Noteworthy, IL-1β has been implicated in perpetuating immune responses and contributing to disease severity in a variety of CNS diseases ranging from diabetic retinopathy, traumatic brain injury, multiple sclerosis, and neurodegeneration.

Neuroinflammation is mainly driven by the activation of glial cells. Specifically, microglia play a critical role in CNS immunomodulation as well as in the clearance of dying neurons, pathogens, and other substances, such as unfolded proteins or cellular debris which are necessary for brain homeostasis [44]. CD16/CD32 is a classic pro-inflammatory marker located in the membrane of microglia and is widely used to detect active microglia [45,46,47]. In our experiments, the expression of CD16/CD32 was increased even with the lowest concentration, showing that the presence of TMAO activates microglia in a pro-inflammatory manner.

Phagocytosis was previously considered to be a beneficial process for homeostasis, preventing the spillover of neurotoxic and pro-inflammatory substances by clearing dying cells [48]. However, nowadays the evidence suggests that microglia activation relies on different targets and receptors that finely control its responses. “Eat-me signals” play a crucial role in the recognition of extracellular debris and initiation of phagocytosis by activating signaling cascades and phagocytic receptors trying to facilitate discriminative clearance of apoptotic cells [49]. For example, phagocytosis of apoptotic neurons has been associated with decreased production of pro-inflammatory cytokines [50], while myelin debris phagocytosis has been shown to enhance the pro-inflammatory phenotype and dampen the anti-inflammatory profile of microglia [51].

In the present study, after TMAO treatment, microglia showed higher rates of myelin phagocytosis, mainly with the highest concentrations. While transient neuroinflammation is beneficial to phagocyte pathogens and cell debris, chronically activated microglia acquire a phagocytic profile characterized by a major release of cytotoxic and pro-inflammatory molecules such as TNF-α. Furthermore, excessive phagocytosis may also be harmful as it could lead to phagoptosis of harmed but viable neurons that may recover with time if they are not phagocytosed [46,52]. Myelin phagocytosis is a process that has been linked to phagoptosis [46]; therefore, excessive myelin phagocytosis as we have found in our in vitro experiments could lead to phagoptosis of stressed but viable cells and promote degeneration processes.

In summary, the present study shows that TMAO treatment favors the differentiation of adipocytes and promotes the expression of pro-inflammatory cytokines in macrophages that could infiltrate adipose tissue in obesity. Moreover, TMAO seems to disrupt the blood–brain barrier as it induced a significant decrease in the expression of occludin in hCMECs and also increased expression of pro-inflammatory cytokines in primary neuronal cultures, induced a pro-inflammatory state in primary microglial cultures, and promoted phagocytosis. Emerging evidence shows that there is a relationship between the biology of cognitive dysfunction risk factors, such as aging or obesity, and the pathophysiology of neurodegenerative diseases. In light of these data obtained in vitro, it can be speculated that risk factors can induce microbiota dysbiosis; moreover, when the impairments in bacterial taxa reach a level in which pro-inflammatory bacteria abundance becomes higher than anti-inflammatory bacteria, bacterial metabolites, such as TMAO, trigger BBB leaking, inducing systemic and CNS inflammation, which results in neuroinflammation and subsequent neurodegeneration. Based on this theory, it is tempting to speculate that restoring gut microbiota via probiotic treatment, fecal microbiota transplantation, or TMAO-decreasing compounds may be a promising therapeutic strategy for neurodegenerative diseases.

## 4. Materials and Methods

### 4.1. In Vitro Cell Lines

#### 4.1.1. Adipocytes (Cell Line 3T3-L1)

3T3-L1 mouse pre-adipocytes (ATCC^®^ CL-173™, Rockville, MD, USA) were cultured at 37 °C in humidified air with 5% CO_2_ in Dulbecco’s Modified Eagle Medium (DMEM; gibco, Life Technologies, Paisley, UK) supplemented with a solution 1% of penicillin–streptomycin (Lonza 17-602E) and 10% of calf bovine serum (CBS).

The cells were grown in a tissue culture flask of 75 cm^2^ (BD Falcon, Franklin Lakes, NJ, USA). The medium was changed every 2–3 days and, when a confluence of 70% was reached, the cells were seeded in 6-well cell culture clusters (Costar, Corning, NY, USA). 3T3-L1 cells present a fibroblast-like morphology, but in the presence of a hormone cocktail, they acquire an adipocyte-like phenotype. When a full confluence was reached in the 6-well cell culture clusters, the medium culture was removed and the differentiating hormone cocktail was added, consisting of DMEM (Gibco, Life Technologies, Paisley, UK) supplemented with a solution of 1% penicillin–streptomycin (Lonza 17-602E), 10% of fetal bovine serum (FBS; gibco Life Technologies, Paisley, UK), 10 µg/mL of insulin, 0.5 mM Isobutylmethylxanthine, and 1 µM dexamethasone.

3T3-L1 cells remained in this medium for 2 days. Next, the medium was changed to DMEM (Gibco, Life Technologies, Paisley, UK) supplemented with a solution of 100 U/mL penicillin–streptomycin (Lonza 17-602E), 10% of fetal bovine serum (FBS; gibco Life Technologies, Paisley, UK), and 10 µg/mL of insulin for 2–4 days to round the adipocytes. Then, the adipocytes were maintained in DMEM (Gibco, Life Technologies, Paisley, UK) supplemented with a solution of 1% penicillin–streptomycin (Lonza 17-602E) and 10% of fetal bovine serum (FBS; gibco Life Technologies, Paisley, UK) without hormones until 70% of cells exhibited the mature adipocyte phenotype. The presence of mature adipocytes was identified via visual analysis using an optical microscope.

#### 4.1.2. Macrophages (Cell Line RAW 264.7)

Murine RAW 264.7 macrophages (ATCC^®^ TIB-71™) were cultured at 37 °C in humidified air with 5% CO_2_ in DMEM (Gibco, Life Technologies, Paisley, UK) supplemented with a solution of 1% penicillin–streptomycin (Lonza 17-602E) and 10% of fetal bovine serum (FBS; gibco Life Technologies, Paisley, UK).

The medium was changed every 2–3 days and cells were split at a confluence of 80%. The cells were grown in a tissue culture flask of 75 cm^2^ (BD Falcon, Franklin Lakes, NJ, USA). All cells cultivated were below passage 12.

#### 4.1.3. Blood–Brain Barrier Cells (Cell Line hCMEC/D3)

Human cardiac microvascular endothelial cells (hCMECs/D3) were cultured at 37 °C in humidified air with 5% CO_2_ in Endothelial Cell Growth Basal Medium (EBM; Lonza, CC-3156) and supplemented with Endothelial Cell Growth Medium SingleQuotsTM Supplements (Lonza, Teknovas, CC-4176) to acquire blood–brain barrier-like characteristics.

The medium was changed every 2–3 days and the cells were split at a confluence of 80%. The cells were grown in a tissue culture flask of 75 cm^2^ (BD Falcon, Franklin Lakes, NJ, USA) previously treated with a mixture of Collagen Type I (Sigma, C3867) and sterile water (1:25). To assure adherence of collagen to the flask, the cells were treated overnight at a temperature of 4–8 °C or for 2–3 h at a temperature of 37 °C. All experiments were performed on cell passage below 8.

### 4.2. Primary Neuronal Cultures

Primary neuronal cultures were derived from the hippocampus and cortex of embryonic day 16 (E16) mice. Brain tissue was processed using glass pipettes until neurons were dissociated. The tissue was then placed in a tube with 2 mL of trypsin for 10 min at 37 °C. Once the tissue had precipitated, the trypsin was taken out. The tissue was further washed with optimen supplemented with glucose and penicillin–streptomycin (Lonza 17-602E) and plated in 6-well plates. After 2 h, the optimen mixture was changed to serum-free neurobasal media with B27 supplement (Invitrogen, Gaithersburg, MD, USA) and 2 mM of L-glutamine on poly-L-lysine-treated (0.1 mg/mL; Sigma) 60 mm dishes. Primary neurons were viable for >3–4 weeks under our culturing conditions.

### 4.3. Primary Microglial Cultures

Primary microglial cultures were obtained from adult mice. Briefly, the cortex was dissected out and cut into pieces (200 µm) with a tissue chopper. The tissue was then transferred to a dish with unsupplemented DMEM. Then, the tissue underwent enzymatic digestion with papain for 30 min at 32 °C with shacking at 200 rpm. Papain was previously prepared by dissolving 20 mg of papain in 6 mL of DMEM -/-. Each hemisphere was digested with 6 mL of papain. Once digested, the tissue was allowed to settle for a period of 2 min. Afterward, the tissue underwent mechanical trituration with a pipette while immersed in 2 mL of DMEM supplemented with 15% FBS and 1% penicillin–streptomycin; then, the supernatant was recovered. This step was repeated 3 times. Once the supernatant was recovered, it was centrifuged at 397 g for 5 min at room temperature and the pellet was resuspended in fresh media (DMEM + 15% FBS + 1% P/S) using filters to filter away long debris. Hoechst dye was used to count cells in the hemocytometer. Finally, the desired number of cells was seeded. After 2 h, the cells were rinsed twice with unsupplemented DMEM (the microglia remain in the wells) and treatments were added to each well.

### 4.4. Differentiation Assay

Visual differentiation of the 3T3-L1 cells treated with different concentrations of TMAO (50, 200, 400, and 700 μM) was evaluated using the differentiation assay.

3T3-L1 cells were seeded in 6-well plates and once a differentiation of 70% was reached they were treated with TMAO at different concentrations (10, 50, 100, 200, 400, 500, 700, and 1000 µM) for 24 h at 37 °C in humidified air with 5% CO_2_. After exposure to treatments, the cells were rinsed with PBS solution and left growing undisturbed in fresh 10% FBS-supplemented DMEM for 2 days.

Mature differentiated cells were counted 48 h post treatment (proliferation) using trypan blue-based specific chambers (Life technologies, C10228) for the Countess^®^ automated cell counter (Life technologies).

### 4.5. Cell Metabolic Activity Assay (MTT)

The MTT assay is used for determining mitochondrial dehydrogenase activity in living cells. MTT, also known as (3-(4,5-dimethylthiazol-2-yl)-2,5-diphenyltetrazolium bromide) tetrazolium is reduced to purple formazan by NADH. This formazan is insoluble in water but soluble in organic solvents like DMSO and can be detected at 540 nm using a spectrophotometer.

When cells reached a confluency of 70–80%, the medium was removed and the treatment (200 μL of TMAO at different concentrations) was added to each well, in ascending concentrations. After a 24 h treatment, cell metabolic activity was checked using the MTT study. 

An amount of 0.5 mg/mL of MTT solution was added (sigma, M5655) to each well. After 2 h of incubation at 37 °C, the medium was removed and the precipitated formazan was dissolved in pure DMSO. Then, spectrophotometric reading was conducted at 540 nm (Spectra MR, Dynex technologies, Denkerdorf, Germay). Survival was calculated as a percentage relative to the control sample.

### 4.6. Immunochemistry

Microglial primary cells incubated in 4-well plates were fixed with 1% paraformaldehyde (PFA) for 10 min; then, 500 μL of 1% PFA was added directly into the wells containing 250 μL of cell medium. After that, media and PFA were removed and 200 μL of cold methanol (−20 °C) was added for 3 min.

The cells were washed 3 times with PBS. Then, primary antibody (Table 1) was added (120 μL/well) and the cells were incubated for 2 h at room temperature.

Later, the cells were washed 3 times with PBS and the secondary antibody (Table 1) was incubated in the dark for 45 min. Finally, the cells were washed again 3 times with PBS and then the cells were flipped and coverslipped with less than 1 drop of Fluorsave Reagent (Merck, Cat. 345789) mixed with 0.5 μL of blue Hoechst.

To guarantee comparable immunostaining, sections were treated together under the same conditions. Fluorescence signals were detected using a confocal microscope LSM 510 Meta (Carl Zeiss, Sttutgart, Germany). The experiment was repeated 4 times (n = 4) and a total of 30 cells were captured and analyzed in each experiment.

### 4.7. Phagocytosis Assay

One hundred μL of rat myelin was mixed with 25 μL of Dil dye (Cat 60010, Lot 981214) and incubated at 37 °C for 5 min. Then, 1 mL of Dulbecco’s PBS was added and the mix was sonicated for approximately 15 s until homogenization was achieved. After that, myelin was centrifuged at 16,000× *g* for 3 min at room temperature. The pellet was suspended in 100 μL of Dulbecco’s PBS water (0.5 mg/mL of myelin). For the cell treatment, 30 μL of myelin was mixed with 720 μL of DMEM + 15% FBS + 1% P/S. 

The cells were previously seeded in 24-well plates and 60 μL of myelin + medium was added to each well and incubated at 37 °C in humidified air with 5% CO_2_ for 24 h. Then, primary microglial cells were stained with goat IBA1; the nucleus was stained with blue Hoechst.

### 4.8. Western Blot

The hCMECs/D3 were rinsed with a PBS solution and collected after a 24 h treatment. Cells were centrifuged at 2000 rpm for 5 min and then the supernatant was removed. The pellets were stored at −80 °C. 

The cells were homogenized in a lysis buffer (NaCl 200 mM, HEPES 100 mM, glycerol 10%, NaF 200 mM, Na_4_P_2_O_7_ 2 mM, EDTA 5 mM, EGTA 1 mM, DTT 2 mM, PMSF 0.5 mM, orthovanadate 1 mM, and NP-40, inhibitors of proteases and inhibitors of phosphatases at 1%) and kept cold in ice for 30 min. Then, the cells were centrifuged at 13,000 rpm for 20 min and the supernatant was collected.

Equal quantities of protein (50 µg/lane) were separated via electrophoresis on 7.5% SDS-polyacrylamide gels and consecutively electrophoretically transferred to nitrocellulose blotting membranes (Cat No: 10600048, Amersham). After blocking with Intercept Blocking Buffer (Part No: 927-70001, LI-COR, USA) or 5% milk in TBST for 1 h at room temperature for binding nonspecific sites, membranes were subjected to immunoblotting with primary antibodies overnight at 4 °C (Table 2).

### 4.9. RNA Extraction and qPCR

Total RNA was extracted using TRI reagent (T9424, Sigma-Aldrich, St. Louis, MO, USA). cDNA was synthesized using a high-capacity cDNA Reverse Transcription kit (4368814, Applied Biosystem, Foster City, CA, USA). The mRNA expressions of GAPDH, IL-6 (Mm00446190_m1), IL-1β (Mm00434228_m1), and TNF-α (Mm00443258_m1) were quantified via TaqMan-based quantitative real-time PCR. Ct values obtained for each gene were referenced to GAPDH (ΔCt) and converted to a linear form using the 2^−ΔΔCt^ term as a value directly proportional to the copy number of the complementary DNA and the initial quantity of mRNA.

### 4.10. Statistical Analysis

Data analyses were carried out using Stata v.14 for Windows and GraphPad Prism 6. The normality was checked using Shapiro–Wilk’s test (*p* > 0.05).

The data are presented using a graph bar showing the mean and the SEM. In cases in which two groups are compared (control vs. TMAO), a two-tailed Student’s *t*-test was used. When more than two conditions were compared (comparisons between the controls and different concentrations of TMAO), one-way ANOVA followed by Tukey’s test were used.

## Figures and Tables

**Figure 1 ijms-24-17557-f001:**
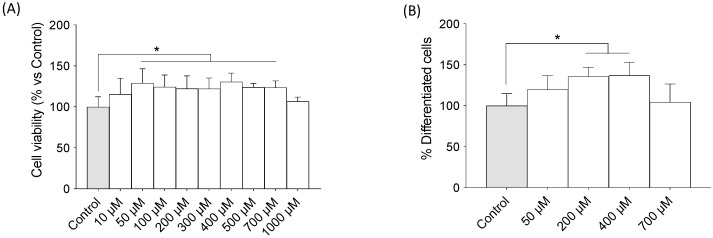
Cell viability and differentiation assay of preadipocytes (3T3-L1) into mature adipocytes. (**A**) The effects of different concentrations of TMAO on the cell viability of 3T3-L1 cells. The cells were treated with TMAO for 24 h. The results are presented as a % of survival and expressed as mean ± SEM (n ≥ 8). One-way ANOVA * *p* < 0.01 for 100, 200, 300, 500, and 700 µM vs. the control; * *p* < 0.001 for 50 and 400 µM vs. the control. (**B**) The effects of different concentrations of TMAO on adipocyte differentiation in 3T3-L1 preadipocytes. The cells were treated with TMAO for 24 h. The results are presented as a % of differentiated cells and expressed as mean ± SEM. Five independent experiments were performed. One-way ANOVA, * *p* < 0.01.

**Figure 2 ijms-24-17557-f002:**
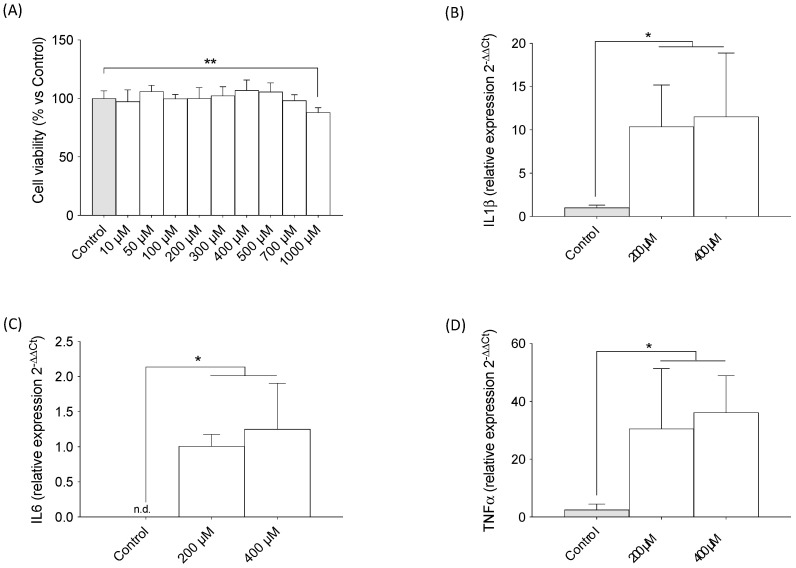
The effects of TMAO on cell viability and pro-inflammatory cytokine expression in macrophages. (**A**) The effects of different concentrations of TMAO on cell viability of macrophages RAW 264.7. The cells were treated with TMAO for 24 h. The results are presented as a % of survival and expressed as mean ± SEM (n ≥ 8). One-way ANOVA ** *p* < 0.01. The effects of different concentrations of TMAO on (**B**) IL-1β (*p* = 0.121), (**C**) IL-6 (*p* = 0.0007), and (**D**) TNF-α (*p* = 0.0038) cytokine expressions. The cells were treated with TMAO for 24 h. The results are presented as a fold change in mRNA expression and expressed as mean ± SEM (n ≥ 4). One-way ANOVA, * *p* < 0.01 for IL-1β and TNF-α; * *p* < 0.001 for IL6. n.d.: non detectable.

**Figure 3 ijms-24-17557-f003:**
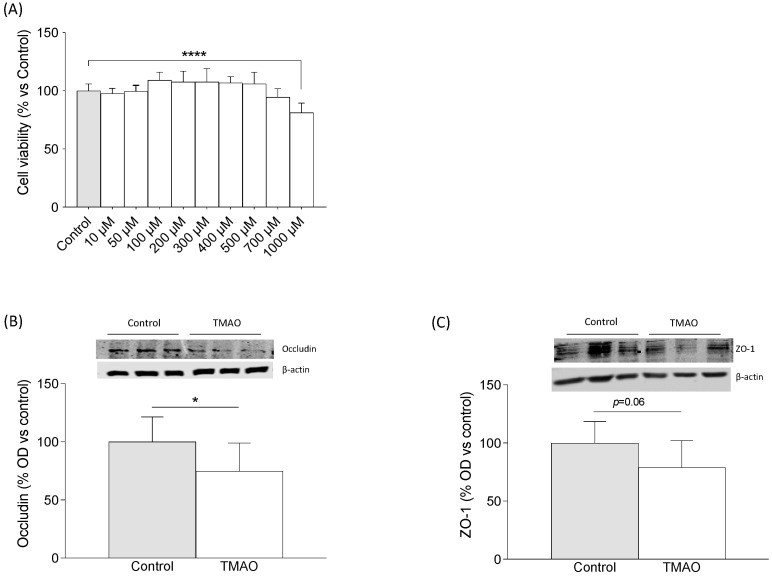
The effects of TMAO on cell viability and tight junction protein expression in hCMECs/D3. (**A**) The effects of different concentrations of TMAO on cell viability of blood–brain barrier cells (hCMECs). The cells were treated with TMAO for 24 h. The results are presented as a % of survival and expressed as mean ± SEM (n ≥ 8). One-way ANOVA **** *p* < 0.0001. The effects of 200 μM of TMAO treatment on (**B**) occludin and (**C**) zonulin expressions in hCMECs/D3. The cells were treated with 200 μM of TMAO for 24 h. The panels show the percentage of optical density (OD) values of the control and representative pictures of the blotting. β-actin was used as the internal loading control. * *p* < 0.05 Student’s *t*-test.

**Figure 4 ijms-24-17557-f004:**
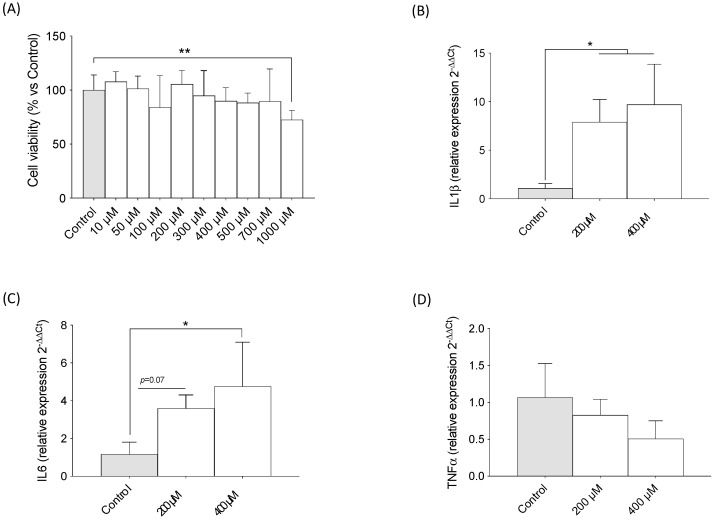
The effects of TMAO on cell viability and pro-inflammatory cytokine expression in primary neuronal cultures. (**A**) The effects of different concentrations of TMAO on cell viability of primary neuronal cell cultures. The cells were treated with TMAO for 24 h. The results are presented as a % of survival and expressed as mean ± SEM (n ≥ 8). One-way ANOVA ** *p* < 0.01. The effects of 200 and 400 μM of TMAO on (**B**) IL-1β, (**C**) IL-6, and (**D**) TNF-α cytokine expressions. The cells were treated with 200 or 400 μM of TMAO for 24 h. The results are presented as a fold change in mRNA expression and expressed as mean ± SEM (n = 4). One-way ANOVA, * *p* < 0.05 for IL-1β 200 µM and IL-6; * *p* < 0.01 for IL-1β 400 µM.

**Figure 5 ijms-24-17557-f005:**
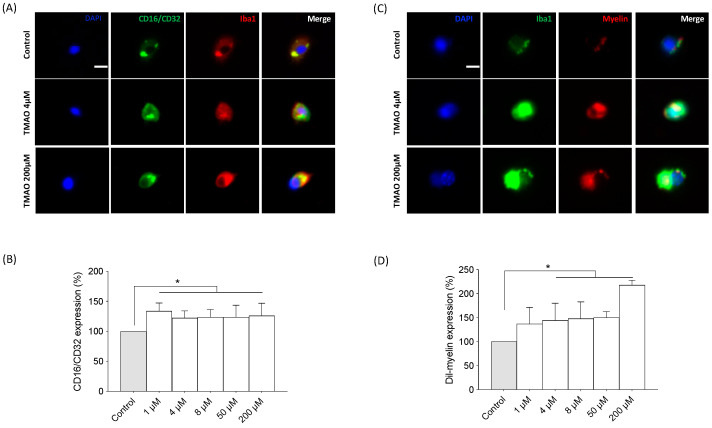
The effects of TMAO on pro-inflammatory activation and phagocytosis in primary microglial cultures. (**A**) Immunohistochemical analysis of the effect of TMAO on the pro-inflammatory marker CD16/CD32 expression in primary microglial cultures and (**B**) quantification of the fluorescence intensity. The cells were treated with TMAO for 24 h. The results are presented as a % of normalized CD16/CD32 expression and expressed as mean ± SEM (n = 4). One-way ANOVA * *p* < 0.05. (**C**) The effect of TMAO on myelin phagocytosis and (**D**) its quantification in primary microglial cultures. The cells were treated with TMAO for 24 h. The results are presented as a % of normalized Dil-myelin expression and expressed as mean ± SEM (n = 4). One-way ANOVA * *p* < 0.01 for 4, 8, and 50 µM vs. the control; * *p* < 0.0001 for 200 µM. Scale bar: 100 µM.

**Table 1 ijms-24-17557-t001:** List of antibodies used for immunofluorescence studies.

Name	Dilution	Reference
IBA-1	1:1000	Abcam (ab5076)
CD16/CD32	1:100	BD Biosciences (553142)
Alexa fluor 546 donkey anti-goat	1:500	Invitrogen
Alexa fluor 488 donkey anti-rat	1:500	Invitrogen
Alexa fluor 488 donkey anti-goat	1:500	Invitrogen

**Table 2 ijms-24-17557-t002:** Primary antibodies used for Western blot experiments.

Name	Dilution	Reference
Ocluddin	(1:1000)	40-4700, Thermo Fisher, MA, USA
ZO-1/TJP1	(1:1000)	40-2200, Invitrogen, MA, USA
p-eNOS at Thr 495	(1:1000)	9574S, Cell signaling, MA, USA
Total eNOS	(1:1000)	9572, Cell signaling, MA, USA
p-SAPK/JNK (Thr 183/Tyr 185)	(1:1000)	9251, Cell signaling, MA, USA
Total SAPK/JNK	(1:1000)	9252S, Cell signaling, MA, USA
GFAP	(1:1000)	3670-S, Cell signaling, MA, USA
CD11b	(1:1000)	NB110-89474, Novus biologicals, Centennial, CO, USA
β-actin	(1:5000)	A1978, Sigma, St. Louis, MO, USA

## Data Availability

The data presented in this study are available upon request from the corresponding author.

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
