# Peer review of "Trimethylamine N-Oxide as a Mediator Linking Peripheral to Central Inflammation: An In Vitro Study"

_ijms, 2023, doi:10.3390/ijms242417557_

Round 1

Reviewer 1 Report

Comments and Suggestions for Authors

Overview of the manuscript

The manuscript focuses on studying of the involvement of Trimethylamine N-oxide (TMAO), a microbiota metabolite, in inducing inflammatory and adipogenic state, as a link between peripheral inflammation and central nervous system inflammation. The authors use several cell line models on which they adopt different molecular and histochemical methodologies. The authors observe that TMAO promoted differentiation of adipocytes, increased expression of proinflammatory cytokines in macrophages, decrease the expression of key membrane factors responsible of blood brain barrier by endothelial cells, increased expression of proinflammatory cytokines in primary neurons, induced a proinflammatory state in primary microglia. The authors conclude that an excessive presence of TMAO derived from a situation of enteral dysbiosis can induce peripheral inflammation, inducing a consequence activation of central nervous system inflammation.

GENERAL  COMMENT

The work analyses an interesting topic related the link between the eubiotic state of the enteric microbiota and the occurrence of inflammation in the central nervous system, as a cause of neurodegenerative conditions. The work is well performed in its experimental plan and the methodological approach is appropriate and sufficient to provide support for the results and conclusions. Bibliographic citations are adequate. Some points of the manuscript should be better addressed. Statistical Analysis paragraph should provide more details.

 Specific comments

 Introduction

Pag. 1, line 42: explain the acronyms.

 Results

The citation “One way ANOVA” into parentheses, should be omitted. You declared the use of the test in Statistical Analysis paragraph.

 Fig. 3: It would be opportune to add to the figure a bar-graph illustrating the OD analysis of Occludin or ZO-1 expression following administration of all TMAO concentrations and related blot.

 Pag. 4, line 140-143: specify at which concentration statistical change is detecetd

 Fig. 4: Insert at lest one blot of the detected inflammatory signals. In the legend you indicated different concentrations, but you only indicate two concentrations. Rephrase the sentence.

 Fig. 5 D: correct the name of the y axis.

 Pag. 6, line 177-182: insulin resistance is not your topic in the present work. Delete the paragraph.

 Pag. 7, line 214:  the sentence is not clear. Rephrase it.

 Pag. 7, line 236: you indicate claudin-5, but you investigated occludin. Correct or explain better the sentence.

 Pag. 7, line 239: the term “disrupting the BBB” is evidence that you do not have. Reduction of occluding and ZO-1 may indicate a probable loss of BBB strength and not destruction. Rephrase the sentence.

 Materials and Methods

Pag. 9, line 312-314: the sentence should be shifted below after the differentiating cocktail administration.

 Pag. 10, line 379: explicit the TMAO concentration administered.

 Pag. 12, Statistical Analysis: Which ANOVA post-hoc test did you used to identify single significance? Indicate where you have used the t-test analysis.

Author Response

GENERAL COMMENT

The work analyses an interesting topic related the link between the eubiotic state of the enteric microbiota and the occurrence of inflammation in the central nervous system, as a cause of neurodegenerative conditions. The work is well performed in its experimental plan and the methodological approach is appropriate and sufficient to provide support for the results and conclusions. Bibliographic citations are adequate. Some points of the manuscript should be better addressed. Statistical Analysis paragraph should provide more details.

We thank the reviewer for his/her positive comments. Statistical analysis section has been extended and clearer stated.

 Specific comments

 Introduction

Pag. 1, line 42: explain the acronyms.

Acronyms have been clarified (please see line 42).

Results

The citation “One way ANOVA” into parentheses, should be omitted. You declared the use of the test in Statistical Analysis paragraph.

As suggested by the reviewer “One way ANOVA” has been deleted in the result section.

Fig. 3: It would be opportune to add to the figure a bar-graph illustrating the OD analysis of Occludin or ZO-1 expression following administration of all TMAO concentrations and related blot.

We apologize if this is not clear enough but while all TMAO concentrations are applied to see how is the cell viability, the study of the impact of TMAO on tight junctions is only performed applying TMAO at a concentration of 200μM. This concentration is carefully chosen as epidemiological studies had showed plasma TMAO in humans can reach 250 μM:

Rio D Del, Zimetti F, Caffarra P, Tassotti M, Bernini F, Brighenti F, et al. Nutrients. 2017;9(10):2–5.

Pag. 4, line 140-143: specify at which concentration statistical change is detected.

As suggested by the reviewer the concentration that induces significant changes has been specified (please see lines 158 and 159).

Fig. 4: Insert at lest one blot of the detected inflammatory signals. In the legend you indicated different concentrations, but you only indicate two concentrations. Rephrase the sentence.

Cytokines changes have been measured by PCR, therefore we cannot add blots in those figures. Regarding the legend, we thank the reviewer to point this into our attention. The sentence has been rephrased (please see line 166).

Fig. 5 D: correct the name of the y axis.

We thank the reviewer to correct this mistake, the axis name has been corrected.

Pag. 6, line 177-182: insulin resistance is not your topic in the present work. Delete the paragraph.

We thank the reviewer to point this issue into our attention. We completely agree with the reviewer that insulin resistance is not the scope of the manuscript, therefore the paragraph has been deleted as suggested by the reviewer.

Pag. 7, line 214:  the sentence is not clear. Rephrase it.

We are sorry not being clear enough. The sentence has been rephrased (please see, line 224-225).

Pag. 7, line 236: you indicate claudin-5, but you investigated occludin. Correct or explain better the sentence.

We are sorry for the mistake; it has been corrected (please see line 246).

Pag. 7, line 239: the term “disrupting the BBB” is evidence that you do not have. Reduction of occluding and ZO-1 may indicate a probable loss of BBB strength and not destruction. Rephrase the sentence.

We completely agree with the reviewer. The sentence has been modified (please see line 249-250).

 Materials and Methods

Pag. 9, line 312-314: the sentence should be shifted below after the differentiating cocktail administration.

We consider that the methods are correct, because cells are first seeded in 6 well plates and then differentiated with the cocktail.

 Pag. 10, line 379: explicit the TMAO concentration administered.

As suggested by the reviewer concentrations have been added (please see line 389-390).

 Pag. 12, Statistical Analysis: Which ANOVA post-hoc test did you used to identify single significance? Indicate where you have used the t-test analysis.

We thank the reviewer to make us specify this, Tukey was used after one-way Anova and now is stated in the methods section.

Reviewer 2 Report

Comments and Suggestions for Authors

Review of  TMAO as a mediator linking peripheral

Overall, this is an interesting investigation and report of the findings. The study is well presented in documenting the effect of TMAO. As the authors point out, there are assumed beneficial and negative aspects with TMAO, as some diets are promoted due to having high level of TMAO. The cell culture approach is a means to differentiate some of the mechanistic  underpinnings in the effect of TMAO on microglia and neurons. The methods are clear and well presented. The results are presented well. It is nice to see  p values reported to closely significance at 0.05 for Figures 3C and 4C as the trends are headed to being significant. The discussion pulls together some aspects to account for whole body issues with adipose tissues and the potential for chronic inflammation which is an area of immense research interest.  

Overall, is an interesting study in defining the underlying mechanisms of TMAO and was well written. However, in the background information, I would potentially add more information about the role of TMAO in the context of gut dysbiosis and make clear that inflammation is prevalent within gut dysbiosis. I think making this connection and defining what dysbiosis is would make contribute well to the background information.

The discussion section is very well written but some of the information (i.e. lines 177-202) could also be better explained within the introduction to provide the reader some more context before going into the results.

There are only some minor suggestions for the authors to consider.

1.     The premise of the paper is that  microbial dysbiosis is related to an increased TMAO secretion.

I would suggest that in addition to many other factors which occur with microbial dysbiosis in the GI tract an additional one is an increased TMAO secretion.

2.     Line 68: The text here reads as a grant proposal.

“ Hence, in this study, we would like to perform in vitro pharmacological and toxicological studies with TMAO to try to understand its role in the organism.”

So maybe write “we performed”

3.     Line 187: There is more than just TMAO with diet to consider. For example, a high fat diets has been shown to lead  to an increase in blood levels of LPS  from the GI microbiome. This can also play substantially to the systemic inflammation. There are many citations which have shown a link to  this associatiosn the authors might consider. As an example:

Dimba NR, Mzimela N, Mosili P, Ngubane PS, Khathi A. Investigating the Association Between Diet-Induced "Leaky Gut" and the Development of Prediabetes. Exp Clin Endocrinol Diabetes. 2023 Nov;131(11):569-576. doi: 10.1055/a-2181-6664. Epub 2023 Sep 26. PMID: 37751850.

So the point is that there is more to consider than just TMAO.

4.     For the readers sake  in the introduction it would be good to explain the benefit of examining for occluding and its importance with tight junction function.

5.     hCMEC/D3 cells . Maybe a reference to past studies which documented these as microvascular Endothelial Cells. But also can be obtained from blood vessels in the CNS as below or some other reference.

Weksler B, Romero IA, Couraud PO. The hCMEC/D3 cell line as a model of the human blood brain barrier. Fluids Barriers CNS. 2013 Mar 26;10(1):16. doi: 10.1186/2045-8118-10-16. PMID: 23531482; PMCID: PMC3623852.

6.     Figure 2B, 2C, and 2D and Figure 3B, 3C, 3D has bolded characters on the Y-axis which looks a little strange formatting-wise. It makes the symbols almost illegible.

Author Response

Overall, this is an interesting investigation and report of the findings. The study is well presented in documenting the effect of TMAO. As the authors point out, there are assumed beneficial and negative aspects with TMAO, as some diets are promoted due to having high level of TMAO. The cell culture approach is a means to differentiate some of the mechanistic  underpinnings in the effect of TMAO on microglia and neurons. The methods are clear and well presented. The results are presented well. It is nice to see  p values reported to closely significance at 0.05 for Figures 3C and 4C as the trends are headed to being significant. The discussion pulls together some aspects to account for whole body issues with adipose tissues and the potential for chronic inflammation which is an area of immense research interest.  

We thank the reviewer for his/her positive comments.

Overall, is an interesting study in defining the underlying mechanisms of TMAO and was well written. However, in the background information, I would potentially add more information about the role of TMAO in the context of gut dysbiosis and make clear that inflammation is prevalent within gut dysbiosis. I think making this connection and defining what dysbiosis is would make contribute well to the background information.

 The discussion section is very well written but some of the information (i.e. lines 177-202) could also be better explained within the introduction to provide the reader some more context before going into the results.

We thank the reviewer this suggestion. A new paragraph has been added in the introduction section that we hope now clarifies better this idea and helps to understand better the discussion afterwards (please see lines 49-55).

There are only some minor suggestions for the authors to consider.

  1. The premise of the paper is that microbial dysbiosis is related to an increased TMAO secretion.

I would suggest that in addition to many other factors which occur with microbial dysbiosis in the GI tract an additional one is an increased TMAO secretion.

 We totally agree with the reviewer. We hope this idea is now clearly explained in the new paragraph that has been added in the introduction (please see lines 49-55) as it states: “Dysbiosis is able to reduce expression of tight junction proteins in the gut, inducing intestinal inflammation and a major permeability, what favors the passage of gut metabolites to the circulatory system (Janeiro et al., 2021). Among these microbial metabolites, trimethylamine N-oxide (TMAO) has been related to inflammation”. With this sentence we try to clearly show that other metabolites apart from TMAO can also be involved.

  1. Line 68: The text here reads as a grant proposal.

“ Hence, in this study, we would like to perform in vitro pharmacological and toxicological studies with TMAO to try to understand its role in the organism.”

 So maybe write “we performed”

As suggested by the reviewer the verb has been changed (please see line 78).

  1. Line 187: There is more than just TMAO with diet to consider. For example, a high fat diets has been shown to lead to an increase in blood levels of LPS from the GI microbiome. This can also play substantially to the systemic inflammation. There are many citations which have shown a link to this associatiosn the authors might consider. As an example:

Dimba NR, Mzimela N, Mosili P, Ngubane PS, Khathi A. Investigating the Association Between Diet-Induced "Leaky Gut" and the Development of Prediabetes. Exp Clin Endocrinol Diabetes. 2023 Nov;131(11):569-576. doi: 10.1055/a-2181-6664. Epub 2023 Sep 26. PMID: 37751850.

 So the point is that there is more to consider than just TMAO.

 We thank the reviewer for bringing this point into our attention. The proposed idea and the suggested reference have been added to the manuscript (please see lines 193-196 and reference 30).

  1. For the readers sake in the introduction it would be good to explain the benefit of examining for occluding and its importance with tight junction function.

 We thank the reviewer for point this issue. A paragraph has been added in the introduction explaining the BBB structure, so it justifies the study of occluding and ZO-1 in our study (please see lines 61-65).

  1. hCMEC/D3 cells. Maybe a reference to past studies which documented these as microvascular Endothelial Cells. But also can be obtained from blood vessels in the CNS as below or some other reference.

 Weksler B, Romero IA, Couraud PO. The hCMEC/D3 cell line as a model of the human blood brain barrier. Fluids Barriers CNS. 2013 Mar 26;10(1):16. doi: 10.1186/2045-8118-10-16. PMID: 23531482; PMCID: PMC3623852.

 We thank the reviewer for giving us the idea and the interesting reference. A paragraph has been added explaining the usefulness of this cell line (please see lanes 135-141) and the reference has been added (please see reference 29).

  1. Figure 2B, 2C, and 2D and Figure 3B, 3C, 3D has bolded characters on the Y-axis which looks a little strange formatting-wise. It makes the symbols almost illegible.

We are sorry for the mistake. All Y axis have been changed so they can be clearly read.